# Explaining socioeconomic inequality in food consumption patterns among households with women of childbearing age in South Africa

**Mweete D. Nglazi**[1,2,3,4]*, **John E. Ataguba**[4,5,6,7]

**1** Implementation Science Centre for Advancing Practice and Training (IMPACT), University of Zambia, Lusaka, Zambia, **2** Department of Clinical Sciences, School of Medicine and Health Sciences, University of Lusaka, Lusaka, Zambia, **3** Department of Community and Family Medicine, School of Public Health, University of Zambia, Lusaka, Zambia, **4** Health Economics Unit, School of Public Health & Family Medicine, University of Cape Town, Cape Town, South Africa, **5** Health Economics Laboratory, Department of Community Health Sciences, Max Rady College of Medicine, Rady Faculty of Health Sciences, University of Manitoba, Winnipeg, Canada, **6** School of Health Systems and Public Health, University of Pretoria, Pretoria, South Africa, **7** Partnership for Economic Policy, Duduville Campus, Kasarani, Nairobi, Kenya

* mweete.nglazi@gmail.com

**Data Availability Statement:** The dataset used for this work are available at the University of Cape

## Abstract

The changing food environment shifts peoples' eating behaviour toward unhealthy food, including ultra-processed food (UPF), leading to detrimental health outcomes like obesity. This study examines changes in socioeconomic inequalities in food consumption spending between 2005/06 and 2010/11 in South African households with women of childbearing age (15 to 49) (WCBA). Data come from the 2005/06 and 2010/11 Income and Expenditure Surveys. The distribution of spending according to the NOVA food classification system groupings (unprocessed or minimally processed foods, processed culinary ingredients, processed and UPF products) was analysed using standard methodologies. Changes in spending inequalities between 2005/06 and 2010/11 were assessed using the concentration index ($C$), while the factors explaining the changes in spending inequalities were identified using the Oaxaca decomposition approach. The Kakwani index ($K$) was used to assess progressivity. Results show that average real spending on all food categories, including UPF, increased between 2005/06 and 2010/11. Socioeconomic inequality in UPF consumption spending decreased ($C = 0.498$ in 2005/06 and $C = 0.432$ in 2010/11), and spending on processed foods ($C = 0.248$ in 2005/06 and $C = 0.209$ in 2010/11). Socioeconomic status, race, and urban residence contributed to overall socioeconomic inequality and changes in UPF consumption inequality between 2005/06 and 2010/11. Spending on all food categories was regressive in 2005/06 ($K = -0.173$ for UPF and $-0.425$ for processed foods) and 2010/11 ($K = -0.192$ for UPF and $-0.418$ for processed foods) because such spending comprises a larger share of poorer household's income than their wealthier counterparts. The government should address these contributors to inequality to mitigate the risks associated with UPF consumption, especially among less affluent households.

Town's DataFirst portal (https://www.datafirst.uct.ac.za/).

**Funding:** The work reported herein was made possible through funding by the South African Medical Research Council (SAMRC) through its Division of Research Capacity Developmenct under the National Health Scholarship Programme from funding received from the Public Health Enhancement Fund/South African National Department of Health. The content hereof is the sole responsibility of the authors and does not necessarily represent the official views of the SAMRC. JEA is supported by the Canada Research Chair. The funders had no role in study design, data collection and analysis, decision to publish, or preparation of the manuscript.

**Competing interests:** The authors have declared that no competing interests exist.

## Introduction

The concept of nutrition transition, developed by Barry Popkin in 1993 [1], is described as the change in diet patterns causing a shift from consuming traditional foods to more processed, high-energy-dense foods of low nutritional value [1–3]. Driven by urbanisation and rapid social and economic development, the nutrition landscape of many countries, particularly low- and middle-income countries, has changed rapidly during the past four decades because of the nutrition transition [4, 5]. Consequently, this unfavourable change in diet patterns has led to a rise in overweight, obesity and diet-related NCDs worldwide [4]. Popkin [1] outlines five patterns of diet and lifestyle: hunter-gatherer lifestyles; early labour-intensive agriculture with periods of famine; receding famine as agriculture becomes more industrialised and incomes rise; Westernised diets high in calories, sugar, animal fat and processed foods, and sedentary lifestyles; and healthier diets and more active lifestyles [1].

Alongside the nutrition transition, two change processes occur in demographic and disease profiles. First, the demographic transition (changes in population dynamics associated with economic growth) causes a gradual decline from high to low fertility and mortality rates [6]. Second, the epidemiologic transition (change in disease patterns and causes of death). As mortality declines during the demographic transition, epidemiological transition causes a shift from acute infectious diseases to chronic degenerative diseases, leading to death and disability [7]. Understanding the nutrition transition is crucial for analysing changes in household consumption patterns over time, particularly in the context of socioeconomic inequality, which significantly influences dietary choices and the prevalence of related diseases [8–10].

The food environment, which is "*the physical, economic, political and socio-cultural context in which consumers engage with the food system to make their decisions about acquiring, preparing and consuming food*" [11, p28], plays a critical role in shaping these dietary choices. The easy accessibility, affordability and availability of processed foods encourage unhealthy eating habits [12]. Changes from traditional diets to processed high, energy-dense foods, especially in Africa, are closely linked with urbanisation and rapid socioeconomic development [2, 3]. As the food environment changes, inequalities in food consumption [13] emerge, driven by the complex interplay of cultural/behavioural, psychosocial and material circumstances. These factors are often referred to as the social determinants of health, including access to adequate, affordable, quality and safe food; food promotion, advertising and information; living and working environments; broader political, socioeconomic, and cultural surroundings; policies; commercial interests; and climate change, among others [11, 14, 15]. The unequal distribution of these social determinants significantly contributes to differences in food consumption and eating behaviour and inequalities in critical health outcomes like obesity [15].

In many developing countries, including South Africa, research on the general adult population shows that the purchase and consumption of healthy food vary by socioeconomic status, with lower socioeconomic groups purchasing or eating fewer fruits and vegetables but more oils and fats than wealthier groups [16–28]. Relatively poorer socioeconomic groups also purchase or eat more ready-made meals, less healthy takeaways or food prepared outside the home [20, 21, 29–34] compared to their richer counterparts. They also eat more sugar-sweetened foods and beverages [35, 36], fewer meat products but more maise meal as reported in Tanzania [37], and more traditional diets comprised of fish and rice as found in Seychelles [23], than their wealthier counterparts.

Socioeconomic patterning also exists in consuming specific food items in developed countries. For example, fruit and vegetable consumption positively correlates with socioeconomic status indicators in Korea and Finland, strengthening the socioeconomic patterning over time [38, 39]. In the United States, weak associations existed between meat consumption and

socioeconomic status but with a U-shaped trend between 1988 and 2004 [40]. In the Netherlands, however, meat and visible fat consumption among the adult population was inversely associated with socioeconomic status, with a stable trend between 1987 and 1998 [21].

Generally, consuming ultra-processed food (UPF), which usually contains artificial colourants, flavours and preservatives and added sugar, salt and fats [41], contributes significantly to the growing non-communicable disease burden, including socioeconomic inequality in overweight and obesity [42–44]. Although differences in the affordability of UPF between high- and low- and middle-income countries [45] may affect results, a negative association existed between socioeconomic status and UPF consumption in Norway, France and the United States [46–48]. However, in Brazil and Chile, a positive association was reported [49–52]. There were no significant differences in the consumption of UPF between income groups in Canada [53].

A recent systematic review investigated the associations between UPF consumption and health outcomes [44] and found no studies reporting an association between UPF consumption and favourable health outcomes. The review revealed that UPF consumption was associated with detrimental health outcomes in 37 of the 43 reviewed studies. In adults, these detrimental outcomes included not only overweight or obesity but a range of conditions such as cardio-metabolic risks, type-2 diabetes, some cancers, cardiovascular diseases, irritable bowel syndrome, depression, frailty condition and all-cause mortality risk [44]. In children and adolescents, these detrimental health outcomes were cardio-metabolic risks and asthma [44].

Data on aggregate food available for consumption in South Africa show a shift in consumption patterns between 1994 and 2015 to a diet with more sugar, sugar-sweetened beverages, more processed and packaged food including edible vegetable oils, increased intake of animal source foods, more added caloric sweeteners, and fewer vegetables [54]. Unfortunately, using aggregate data describing per capita instead of actual food consumption and spending at the household or individual level is insufficient to adequately show the patterns or assess socioeconomic inequalities in food consumption [54–56]. Another study in South Africa found high levels (39.4%) of UPF consumption among low-income men and women aged 18 to 50 years living in two of the nine provinces [57]. Given the high burden of diseases associated with consuming UPF and other unhealthy diets and the significant burden of overweight and obesity among women in South Africa, this paper begins by examining the distribution of household spending according to the NOVA food classification system groupings (unprocessed or minimally processed foods, processed culinary ingredients, processed and UPF products). It also assesses the changing pattern in socioeconomic inequalities in UPF consumption and the progressivity of these expenditures between 2005/06 and 2010/11 in South African households with women aged 15 to 49 years. This paper also assesses the factors that explain changes in socioeconomic inequality in UPF consumption in South Africa. To the best of our knowledge, this represents the first of such analyses in South Africa and Africa, focusing on households with women aged 15 to 49 years, a population with high overweight and obesity prevalence [58].

## Methods

### Ethics statement

This paper uses publicly available Income and Expenditure Survey data that have received ethics approval. IES is a routine national survey conducted by Statistics South Africa, the national statistical authority. Although secondary data are used in this paper, ethics approval was also received from the Human Research Ethics Committee at the University of Cape Town (HREC Reference 9/2019).

## Data sources

Ideally, individual-level food consumption or expenditure data are needed to investigate food consumption inequality. However, these data are expensive to collect and are often unavailable at the nationally representative scale in South Africa and Africa broadly. Thus, this paper uses consumption per capita from household data, assuming homogeneity in food consumption patterns among individuals within each household. Data come from two rounds (2005/06 and 2010/11) of the nationally-representative Income and Expenditure Survey (IES) conducted by Statistics South Africa (Stats SA). The 2005/06 IES fieldwork was between September 2005 and August 2006. A total of 3,000 primary sampling units were selected, with 22,617 households sampled, corresponding to a 93.5% response rate [59]. Fieldwork for the 2010/11 IES was between September 2010 and August 2011, with 27,665 households (91.4% response rate) sampled from 3,254 primary sampling units [60]. The diary and questionnaire recall methods were used in both IES rounds to collect data; households were visited five times—one for the main questionnaire and four for the weekly diaries [59, 60]. The IES 2005/06 and 2010/11 rounds are comparable, but previous rounds (1995 and 2000) used face-to-face recall only with a single household visit.

## Food classification according to the NOVA system

Food and non-alcoholic beverages reported in the IES were classified according to the NOVA food classification system based on the level of industrial food processing [45, 61]. The NOVA system is commonly used in public health nutrition research and policy [62] and has four food groups—i) Unprocessed or minimally processed foods include fruits, vegetables, grains or cereals, potatoes and tubers, pulses (dried beans, peas and lentils), meat (beef, pork, mutton and other meat products), fish, dairy, eggs, unsalted nuts and seeds, dried herbs, coffee and tea. ii) Processed culinary ingredients include fats and oils, table sugar, flours, pasta, honey and table salt. iii) Processed food products include tinned vegetables, tinned legumes or fruits, salted nuts or seeds, processed meat or fish, tinned fish, cheese, and bread. iv) UPF products include confectionary (ice cream, chocolate, sweets), sugar-sweetened beverages, snacks, breakfast cereals and baked goods. Details of the four NOVA groups are contained in S1 Table.

## Key variables

Table 1 contains a description of the key variables used in this paper. The variable selection was based on the broader determinants of health [14] and their availability in the IES datasets. Household consumption expenditure was used as a measure of socioeconomic status. Household consumption expenditure and spending according to the NOVA food classification system were divided by household size to generate per capita variables. The expenditure data for 2005/06 and 2010/11 were adjusted to 2016 prices using the consumer price index to obtain real values. It is important to note that the paper focuses on households with women of childbearing age (WCBA) aged between 15 and 49 years.

## Analytical methods

**Concentration index.** Socioeconomic inequality in food spending by the NOVA groups was assessed using the concentration index, $C$ [64]. For simplicity, the standard $C$ is computed via the regression equation [65]:

$$2\sigma_r^2\left(\frac{y_i}{\mu}\right) = \alpha + Cr_i + \epsilon_i \tag{1}$$

**Table 1. Description of key variables.**

| Expenditure variables | |
|---|---|
| Unprocessed or minimally processed foods (per capita) | Total household spending on unprocessed or minimally processed foods divided by household size. |
| Processed culinary ingredients (per capita) | Total household spending on processed culinary ingredients divided by household size. |
| Processed food products (per capita) | Total household spending on processed food products divided by household size. |
| Ultra-processed food products (per capita) | Total household spending on ultra-processed food products divided by household size. |
| Total expenditure on food and non-alcoholic beverages (per capita) | Total household spending on food and non-alcoholic beverages divided by household size. |
| Total household consumption expenditure (per capita) | Total household spending on cost of housing, food, non-alcoholic beverages, alcoholic beverages, clothing, footwear, health services, recreation and entertainment and own consumption of homegrown products divided by household size. |
| **Determinants** | |
| Population group | Black African[1] = 1 for household head self-identified as black African race; 0 otherwise. |
| | Coloured = 1 for household head self-identified as coloured; 0 otherwise. |
| | Indian/Asian = 1 for household head self-identified as Indian/Asian race; 0 otherwise. |
| | White = 1 for household head self-identified as white; 0 otherwise. |
| Area of residence[2] | Urban = 0 if a household is in a rural location. |
| | Urban = 1 if a household is in an urban location. |
| Quintiles of socioeconomic status (Quintiles 1–5)[3] | Quintile 1 = 1 if a household is in the poorest socioeconomic group; 0 otherwise. |
| | Quintile 2 = 1 if a household is in the second poorest socioeconomic group; 0 otherwise. |
| | Quintile 3 = 1 if a household is in the middle socioeconomic group; 0 otherwise. |
| | Quintile 4 = 1 if a household is in the second richest socioeconomic group; 0 otherwise. Quintile 5 = 1 if a household is in the richest socioeconomic group; 0 otherwise. |

*Notes*:

[1]The South African population is predominantly black [63];

[2]Urban residence refers to households in a big city (metro), town or township, while rural residence refers to rural or traditional areas. Enumerators were trained to record the location of each household;

[3]Quintiles of socioeconomic status, which groups the population into five quintiles, each containing approximately 20% of the population, are based on household consumption expenditure per capita.

where $\mu$ is the mean of food spending ($y_i$), $\sigma_r^2$ is the variance of the fractional rank ($r_i$) of the socioeconomic status, $\alpha$ is the intercept, and $\epsilon_i$ is the error term [65, 66]. Consumption expenditure per capita was used to measure socioeconomic status, which was obtained by dividing total consumption expenditure, adjusted to 2016 prices at the household level, by the total household size.

The value of the *C* ranges from -1 to +1. A positive concentration index ($C > 0$) means that food spending in the specific NOVA group is more likely among wealthier than poorer households with women aged 15–49 years, while a negative index ($C < 0$) signifies the opposite.

**Decomposing the concentration index.** The $C$ is decomposed to explain the drivers (that is, the contributions of the determinants listed in Table 1) of socioeconomic inequalities in food spending. Here, only the concentration index in UPF spending is decomposed as this is the major driver of many adverse health outcomes, including overweight and obesity. Consider the relationship between UPF spending ($y$) and the determinants ($x$):

$$y_i = \alpha + \sum_k \beta_k x_{k_i} + \varepsilon_i \tag{2}$$

where $\alpha$ and $\beta$ are ordinary least squares parameters, and $\varepsilon$ denotes the error term.

Wagstaff, Van Doorslaer [67] show that the concentration index for $y$ (that is, $C$), can be rewritten taking into account the relationship in Eq 3 as follows:

$$C = \underbrace{\sum_k \left( \frac{\beta_k \bar{x}_{k_i}}{\mu} \right) C_k}_{\text{explained}} + \underbrace{\frac{GC_\varepsilon}{\mu}}_{\text{unexplained}} \tag{3}$$

where $\mu$ remains the mean of UPF spending, $\bar{x}$ is the mean of each regressor, $\beta_k$ is the ordinary least squares coefficient for each of the explanatory factors from Eq 3, $C_k$ denotes the concentration index for the $k$-th contributing factor, while $GC_\varepsilon$ is the generalised concentration index for the error term ($\varepsilon$) in Eq 2 [67]. The explained component in Eq 3, which represents the contribution of each determinant to the concentration index, is the product of the elasticity of $y$ with respect to each determinant $\left( \frac{\beta_k \bar{x}_{k_i}}{\mu} \right)$ and $C_k$. With a pro-rich $C$ where $C > 0$, a positive sign on a contributing factor denotes that, ceteris paribus, that factor contributes to the concentration of UPF spending among wealthier households or the socioeconomic inequality in UPF spending would be reduced if that factor was not present (that is, $\left( \frac{\beta_k \bar{x}_{k_i}}{\mu} \right) = 0$ or $C_k = 0$). The opposite applies for a negative sign on a contributing factor.

**Decomposing changes in the concentration index.** The Oaxaca decomposition method [67] was used to assess and explore factors contributing to socioeconomic inequalities in UPF spending changes between 2005/06 and 2010/11. The decomposition was based on UPF product spending only, as these are products linked to obesity. Another useful decomposition framework proposed by Ataguba, Nwosu [68] focuses on decomposing changes in socioeconomic inequality into what the authors called between- and within-socioeconomic group inequality. Unfortunately, this was not applied in this paper as it does not provide factors explaining changes in inequality between two time periods.

The Oaxaca decomposition approach [69] is performed using the formula below:

$$\Delta C = \sum_k \eta_{kt}(C_{kt} - C_{kt-1}) + \sum_k C_{kt-1}(\eta_{kt} - \eta_{kt-1}) + \Delta \left( \frac{GC_{\varepsilon t}}{\mu_t} \right) \tag{4}$$

where $t$ is the time period corresponding to 2010/11, and $t-1$ corresponds to 2005/06. $\Delta$ denotes the first differences and $\eta_k$ is the elasticity of $y$ with respect to $\bar{x}_k$, as in Eq 4. The Oaxaca method allows for the change in socioeconomic inequality in UPF spending over time to be decomposed into changes in the contributing factors of the concentration index and elasticities of the determinants of UPF spending. However, one limitation of the method is that it is challenging to disentangle changes within elasticities [66].

**Assessing progressivity in food spending by NOVA classification.** Progressivity was assessed using two approaches. The first involved comparing the share of total household expenditure spent on food by quintiles of household consumption expenditure for each

NOVA food classification group [70]. If the share increases with quintiles, expenditures on the NOVA food group account for a larger share of total household consumption expenditures for wealthier than their less affluent counterparts, and spending on the NOVA food category is progressive. The spending is regressive when the share decreases with quintiles. The second approach uses the Kakwani index of progressivity that compares the distribution of household consumption expenditure using the Lorenz curve (or the Gini index) with that of food spending using the concentration curve (or the concentration index) [70].

In this paper, the Kakwani index ($K_j$) for each NOVA food group ($j$) was calculated as the difference between the Gini index ($G$) of consumption expenditure and the concentration index ($C_j$) of expenditure on the particular NOVA food group:

$$K_j = C_j - G \tag{5}$$

Progressive spending on the NOVA food group ($j$) occurs when $K_j > 0$ or $C_j > G$, whereas regressive spending occurs when $K_j < 0$ or $C_j < G$ and proportional spending occurs when $C_j = G$. The values of $K_j$ range from -2 (most regressive) to 1 (most progressive) [66]. Also, changes in the progressivity or regressivity of food spending were assessed as outlined in Ataguba [71], which could be a pro-poor, pro-rich, or proportional shift.

All analyses were done in Stata [72], accounting for the IES sampling structure. The standard errors for the various components of the concentration index decomposition in Eq 4 were obtained using a bootstrapping procedure with 250 resamples based on the sampling structure [73].

## Results

In 2005/06, 16,209 (76.7%) households had at least one woman aged 15–49 years; this was 17,217 (68.0%) in 2010/11. Among households with at least one woman aged 15–49 years, nearly half were female-headed households, most household heads self-identified with the Black African population group, and almost two-thirds were in urban areas in both 2005/06 and 2010/11 (see S2 Table).

Per capita spending on food and non-alcoholic beverages increased between 2005/06 and 2010/11, with wealthier households spending more on food and non-alcoholic beverages than poorer households, as expected (Table 2). Average annual per capita spending on unprocessed or minimally processed foods, processed culinary ingredients, processed foods, and UPF products increased by US$61.54, US$22.26, US$66.58 and US$ 37.24, respectively, in real terms between 2005/06 and 2010/11 (Table 2).

Socioeconomic inequality in food expenditures is shown in Table 3, using the concentration index. Based on the results in Table 2, and as expected, the concentration indices in

**Table 2. Average annual per capita consumption (in US$) on specific NOVA food classification in South African households with women aged 15–49 years in real terms, 2005/06–2010/11.**

|  | Total food and non-alcoholic beverages | | Unprocessed or minimally processed foods | | Processed culinary ingredients | | Processed food products | | Ultra-processed food products | |
|---|---|---|---|---|---|---|---|---|---|---|
|  | 2005/06 | 2010/11 | 2005/06 | 2010/11 | 2005/06 | 2010/11 | 2005/06 | 2010/11 | 2005/06 | 2010/11 |
| Poorest quintile | 126.76 | 195.13 | 57.01 | 96.13 | 17.58 | 37.23 | 51.63 | 103.06 | 16.57 | 34.54 |
| 2nd quintile | 183.91 | 289.46 | 80.16 | 131.26 | 19.72 | 41.05 | 66.89 | 128.37 | 26.37 | 48.20 |
| 3rd quintile | 224.31 | 369.84 | 97.43 | 163.90 | 21.28 | 44.26 | 74.71 | 146.31 | 35.96 | 71.13 |
| 4th quintile | 284.41 | 418.34 | 129.03 | 198.69 | 21.04 | 42.73 | 84.75 | 158.00 | 53.68 | 97.18 |
| Richest quintile | 575.97 | 693.73 | 240.70 | 334.82 | 28.03 | 53.29 | 125.21 | 200.83 | 141.06 | 203.74 |
| **Total** | **303.66** | **407.30** | **131.33** | **192.88** | **22.06** | **44.32** | **84.50** | **151.08** | **62.32** | **99.56** |

**Table 3. Progressivity of spending on specific NOVA food classification groups in South African households with women aged 15–49 years, 2005/06–2010/11.**

| NOVA Food classification | Gini index | Concentration index | Kakwani index | Gini index | Concentration index | Kakwani index |
|---|---|---|---|---|---|---|
| | | 2005/06 | | | 2010/11 | |
| Total food and non-alcoholic beverages | 0.673*** (0.008) | 0.377*** (0.007) | -0.296*** (0.020) | 0.629*** (0.008) | 0.319*** (0.006) | -0.309*** (-0.019) |
| Unprocessed or minimally processed foods | 0.673*** (0.008) | 0.370*** (0.006) | -0.303*** (0.020) | 0.628*** (0.008) | 0.343*** (0.005) | -0.285*** (0.019) |
| Processed culinary ingredients | 0.673*** (0.009) | 0.187*** (0.007) | -0.486*** (0.022) | 0.627*** (0.009) | 0.169*** (0.006) | -0.458*** (0.022) |
| Processed food products | 0.673*** (0.009) | 0.248*** (0.006) | -0.425*** (0.022) | 0.627*** (0.009) | 0.209*** (0.005) | -0.418*** (0.021) |
| Ultra-processed food products | 0.671*** (0.009) | 0.498*** (0.007) | -0.173*** (0.020) | 0.624*** (0.009) | 0.432*** (0.007) | -0.192*** (0.022) |

*Notes*:

(1) Significance levels are as follows: *** p< 0.01, ** p< 0.05, *p< 0.10.

(2) Standard error displayed in parentheses.

(3) Gini indices are slightly different because only complete data pairs (food consumption and overall consumption) are used for each computation

Table 3 are positive and statistically different from zero (that is, pro-rich), meaning that wealthier households with at least one woman aged between 15 and 49 years spend more on the various NOVA food categories than poorer households in absolute terms.

Between 2005/06 and 2010/11, the statistically significant pro-rich socioeconomic inequality in UPF consumption, using the concentration index, decreased from 0.50 to 0.43 (Table 3). The decomposition of the concentration index of UPF spending in Fig 1 shows that in 2005/06, socioeconomic status (about +30%), race (about +29%) and urban residence (about +8%) contributed positively to socioeconomic inequality in UPF spending. The residual contributed approximately 34% of the total socioeconomic inequality in UPF product spending. The 2010/11 results were similar to those for 2005/06 (Fig 1), with urban residency becoming less prominent.

The Oaxaca decomposition in Fig 1 shows that socioeconomic status (about +37%) is the largest contributor to changes in socioeconomic inequality in UPF spending between 2005/06 and 2010/11. Also, race (approximately +30%) and urban residency (approximately +19%) contributed to the decline in the concentration index of UPF spending from 0.50 to 0.43 (that is, becoming less pro-rich) between 2005/06 and 2010/11. The detailed results of the contributions of the different factors to socioeconomic inequality in spending on ultra-processed food products in South African households with women aged 15 to 49 years are presented in S3 Table.

The preliminary progressivity results in Fig 2 indicate that wealthier households (that is, richer quintiles) with at least a woman aged between 15 and 49 years spend a smaller share of their expenditure on the various NOVA food groups than their poorer counterparts (that is, poorer quintiles) in both 2005/06 and 2010/11. In other words, NOVA food categories account for a larger share of total expenditure by poor households than wealthier households. This relationship means expenditures on unprocessed or minimally processed foods, processed culinary ingredients, processed foods, and UPF products were regressive in 2005/06 and 2010/11. The same conclusion of regressive spending on the various NOVA food groups is reached using the Kakwani index, as shown in Table 3. The Kakwani index results indicate that spending on most NOVA food groups became less regressive between 2005/06 and 2010/11, except for UPF and total food and non-alcoholic beverages spending. There was a slight increase in

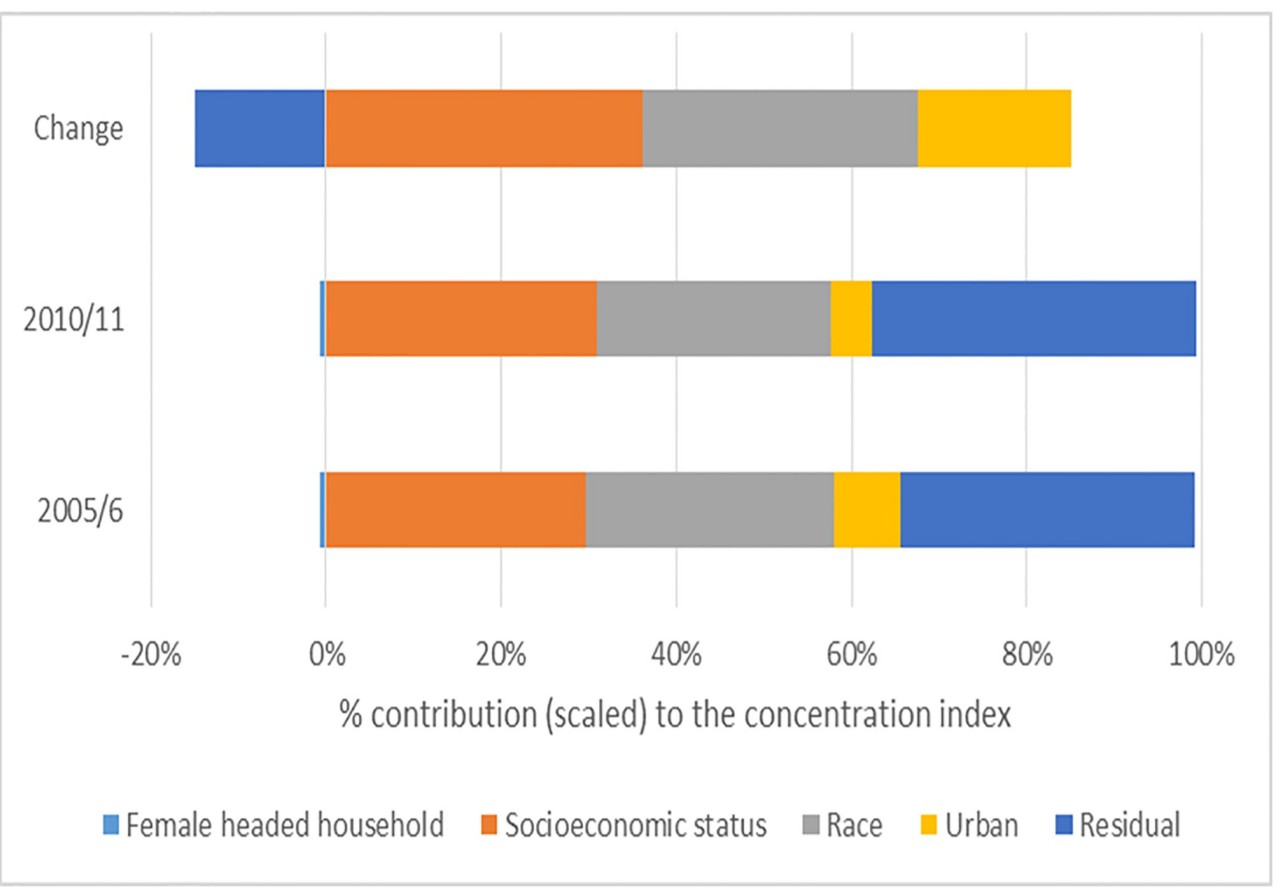

**Fig 1. Contributions of determinants on spending on ultra-processed food products in South African households with women aged 15 to 49 years, 2005/6, 2010/11 and between the years.**

UPF spending regressivity, with the Kakwani index decreasing from -0.173 (in 2005/06) to -0.192 (in 2010/11).

## Discussion

Although average spending on selected foods (unprocessed or minimally processed foods, processed culinary ingredients, processed food and UPF products) increased in real terms, socioeconomic inequality in the consumption of UPF reduced between 2005/06 and 2010/11, albeit with consistently positive concentration indices in households with at least a woman aged between 15 and 49 years. The decomposition analyses suggest that socioeconomic status and race were major contributors to socioeconomic inequality in UPF consumption, with socioeconomic status, race and urban residency contributing to reducing the pro-rich socioeconomic inequality in UPF spending between 2005/06 and 2010/11. This paper also found that wealthier households with women aged 15–49 years spend a relatively smaller share of their expenditure on each specific NOVA food group than poorer households, making such spending regressive. However, spending on most NOVA food groups became less regressive between 2005/06 and 2010/11.

The finding that socioeconomic inequality in UPF product consumption was pro-rich is similar to results from other studies from Brazil and Chile, which found a positive association

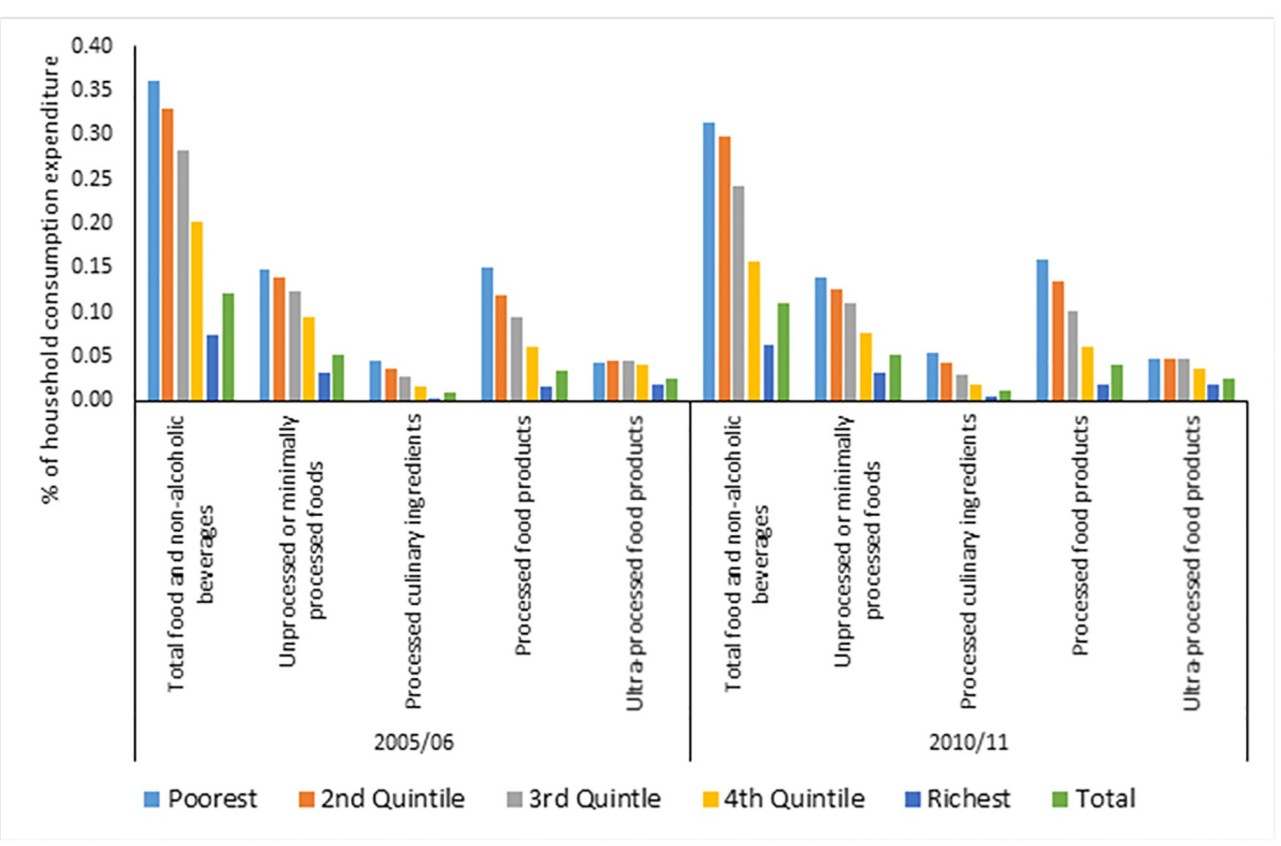

**Fig 2. Share of total household consumption expenditure spent on specific NOVA food classification groups in South African households with women aged 15–49 years, 2005/06–2010/11.**

between socioeconomic status and UPF consumption [49–52]. However, other studies from Norway, France and the United States showed a negative association between socioeconomic status and UPF consumption [46–48]. The differences in UPF product consumption between income groups in Canada were similar [53]. While differences in UPF products' affordability may explain the differences in the results across countries, in South Africa, food consumption patterns have changed significantly in the past decades toward Western-oriented diets that are readily available and affordable to the rich [54]. Diet choice forms part of individuals' strategies to position into higher socioeconomic groups and socially distance themselves from lower-status groups [74]. It could explain the significant concentration of spending on UPF among the rich in South Africa. For example, there is evidence that higher education graduates with better-paying jobs purchase more expensive but not always healthier food products since food might be seen as a status symbol [75].

The socioeconomic inequality in UPF spending, including the decreasing inequalities in UPF spending, was substantially attributed to socioeconomic status and self-classified race. In South Africa, apart from the advantage created by income for the rich, the shift to Westernised diets [54] and lifestyles, especially in Black African communities [76], could explain the significant disparities in socioeconomic inequality in UPF spending. For the wealthy, it may well be that purchasing UPF products, especially foreign products, is perceived as a sign of prestige and status [75, 77]. Urban residency positively reduced the pro-rich socioeconomic inequality in UPF product consumption in South Africa. Without disparities in UPF consumption

between urban and rural areas, the concentration index of UPF consumption would have been more pro-rich. Studies from elsewhere have shown that UPF product consumption is higher in urban areas [49, 78, 79], households with women working outside the home [79], households with time constraints, households with many children, and individuals living with overweight or obesity [47] than among their counterparts. A substantial portion of the socioeconomic inequality was not explained by the factors included in the model, suggesting that unobserved factors, including individual-level determinants and unquantifiable idiosyncrasies, drive consumption patterns towards UPF. For example, household consumption patterns might differ from individual consumption patterns, depending on autonomy and decision-making abilities. Unfortunately, these factors are not included directly in the model but may be contained in the unexplained component.

Changes in food consumption patterns due to changes in disposable income are well-known in the consumer theory literature. Food Engel Curve expenditures are tools to understand food consumption patterns and household welfare [80, 81]. While many studies have been conducted in developed countries, there is scant literature in developing countries. A study in another developing country analysed food consumption patterns in Rwandan households using nationally representative survey data [80]. The study findings indicated that poor households spend more on food that contains higher carbohydrates and starches. Furthermore, the study found that many rural households spend less on micronutrients from animal products [80].

This study has strengths and limitations. The study's strength is that comparable nationally representative data (IES 2005/06 and IES 2010/11) are used. A limitation of the study is that the household consumption pattern was assumed to be congruent with the consumption pattern for women aged 15–49 years in the household. While this alignment may not always be the case, as consumption patterns are likely heterogeneous and differ by age and sex, there is likely a substantial correlation between individual and household consumption patterns for a large portion of the population, more so for the less wealthy. We note that further research is needed using data at the individual level that focuses on women of childbearing age 15–49 years to address the assumption of homogeneity in food consumption patterns among individuals within each household. Also, due to data limitations, household-level determinants were used for the decomposition analysis mainly because the household, not all women aged between 15 and 49 years, was assumed as the decision-making unit for consumption and food spending. Again, further research using nationally representative women-level (or individual) data could provide more insights into assessing socioeconomic inequalities in food consumption and investigate other determinants, including individual-level determinants, contributing to socioeconomic inequality.

This study has policy implications. One of the chief contributors to socioeconomic inequality in UPF consumption was socioeconomic status. Therefore, the government needs to prioritise households with women aged 15–49 years from different socioeconomic backgrounds, mainly those prone to consuming UPF products and those bearing the disease burden associated with consuming UPF. This approach will substantially reduce socioeconomic inequality in UPF consumption and the burden of non-communicable diseases, including overweight, obesity, cardiovascular disease, diabetes and some cancers. Currently, UPF consumption accounts for a substantial share of poorer than wealthier households' expenditures (Fig 2), which can be reduced significantly with policy to shift consumption towards healthier alternatives, including fresh fruits, vegetables and less processed food products. Cost-effective interventions for achieving this and tackling overweight and obesity include continuing to zero-rate healthier food items from the country's value-added tax [82], while other commodities maintain the 15% value-added tax rate. Unfortunately, apart from the April 2018 increase in

the value-added tax rate from 14% to 15%, it has been over 20 years since the country changed the value-added tax structure, including adding more items for zero-rating [82]. This paper's findings make it imperative for continuous policy reforms and understanding the changing landscape of food consumption, especially for those who are less healthy, which contributes to the growing burden of non-communicable diseases. Taxing unhealthier food products like sugar-sweetened beverages [83] is another complementary policy, including promoting appropriate food labelling and regulating food formulation [84]. Understandably, substantial political will and investments are required in the face of opposition from the food industry to curb the rise in the consumption of UPF products, especially among the deprived population group, including women aged 15–49 years, used in this study. The interests of the food industry that contribute to the rise in obesity are linked to the abundant supply of UPF products (the commercial determinants of health), which need to be regulated [85]. From the individual's perspective, and as noted within the National Strategic Plan for the Prevention of Non-communicable Diseases in South Africa [86, 87], continuous education is needed about healthy lifestyles and being physically active through awareness-raising campaigns and programmes aimed at behavioural changes. A combination of actions and interventions (both demand- and supply-side) will be necessary to reduce the consumption of UPF and the burden of disease associated with UPF consumption in South Africa.

It is critical to note that a combination of access factors influences the consumption of different food types, including food availability, affordability and acceptability [88]. While some individuals may be well aware of the importance of consuming healthy food items, they may be constrained by the availability or affordability of such items. For instance, it may be likely that fast foods are readily available at workplaces where women work, and these women may be constrained by time to prepare homemade food. Overall, looking at both the demand- and supply-side aspects and sensitising people about the importance of healthy food consumption, reductions in the burden of overweight and obesity cannot be achieved without paying attention to these key access and social determining factors identified in this paper.

## Conclusion

In marking World Obesity Day 2022, the World Health Organization, acknowledging the detrimental impact of obesity globally, called for countries to work together towards creating a better food environment where everyone can afford and access healthy diets [89]. In South Africa, this paper finds that spending on UPF, for instance, continues to comprise a significant proportion of poorer households' aggregate expenditure than their wealthier counterparts, with key determining factors explaining the concentration of such spending among poorer households. Although the food environment in South Africa may be changing, socioeconomic status, race, and urban residency remain critical factors to consider in designing, targeting and encouraging healthy eating behaviour. Ongoing sensitisation campaigns need to be conducted on the links between UPF product consumption and the risks of overweight and obesity and how to reduce them. Finally, for policy, we highlight the centrality of prioritising low-income households and those vulnerable to consuming UPF and other less healthy food alternatives in South Africa.

## Supporting information

**S1 Table. NOVA food classification system\* based on the nature, extent and purpose of industrial foods processing.**
(DOCX)

**S2 Table. Descriptive statistics of households with at least one woman aged 15–49 years in South Africa, 2005/06 and 2010/11.**
(DOCX)

**S3 Table. Decomposition of concentration index for spending of ultra-processed food products in South African households with women aged 15 to 49 years, 2005/06, 2010/11 and between the year.**
(DOCX)

## Author Contributions

**Conceptualization:** Mweete D. Nglazi.

**Data curation:** Mweete D. Nglazi.

**Formal analysis:** Mweete D. Nglazi, John E. Ataguba.

**Methodology:** Mweete D. Nglazi, John E. Ataguba.

**Visualization:** Mweete D. Nglazi, John E. Ataguba.

**Writing – original draft:** Mweete D. Nglazi.

**Writing – review & editing:** Mweete D. Nglazi, John E. Ataguba.

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
