## [Decision Letter · Decision Letter 0]

15 Apr 2024

PGPH-D-24-00301

Fighting overweight and obesity in households with women of childbearing age through food consumption: explaining socioeconomic inequality in food consumption patterns in South Africa

Dear Dr. Nglazi,

Thank you for submitting your manuscript to PLOS Global Public Health. After careful consideration, we feel that it has merit but does not fully meet PLOS Global Public Health’s publication criteria as it currently stands. Therefore, we invite you to submit a revised version of the manuscript that addresses the points raised during the review process.

We look forward to receiving your revised manuscript.

Kind regards,

Fabrizio Ferretti, Ph.D.

Guest Editor

Manuscript title: Fighting overweight and obesity in households with women of childbearing age through food consumption: explaining socioeconomic inequality in food consumption patterns in South Africa

Journal: PLOS Global Public Health

Manuscript ID: PGPH-D-24-00301 – March 2024

#1 Reviewer’s Report

Major revision

Major revision comments

01. The title does not fit the manuscript. It suggests that this was an intervention study which is not. The study examines changes in socioeconomic inequalities in food consumption between two surveys and not any intervention to fight overweight and obesity.

02.The main analysis which used Oaxaca decomposition analyzed only UPF product spending which the authors claim is linked to obesity, yet the title mentions both overweight and obesity. Please explain this mismatch.

03. Who was overweight and obese in the study? No definition for this classification has been provided.

04. The authors need to explain and explore the assumption of homogeneity in 5 food consumption patterns among individuals within each household. Although the study only included women aged 15 to 49, most of the food consumption indices are calculated using the household size. Food consumption patter may be different for sex (male vs female) and age, for example infants <6 months should be exclusively breastfed.

05. Table 1 show the major key variables. Most of these variables have not been defined or explained, for example how was rural and urban residence defined? How were the socioeconomic quintiles calculated? Which household assets were included?

#2 Reviewer’s Report

Minor revision

The manuscript addresses a major public health issue affecting South African population. this is a very useful well conducted study that will go along way in assisting with formulation of workable strategies to curb overweight and obesity in South Africa and beyond. I have a few minor correction that need attention to improving the quality of the manuscript.

Minor revision comments

01. Pg 5 line 6. The acronym WCBA needs to be explained further.

02. Pg 11 line 15 and elswhere, the use of the abbreviation "i.e." should be discouraged and use the word itself.

03. Page 27 line 1 and elsewhere, organizations such as WHO should include their location when referencing.

03. Pg 27 line 42 and elsewhere, be consistent when writing journal names in the reference list by capitalising first letter in each word of the name or not doing it at all.

Editor

Recommendation: Major revision

Main review questions

1. Originality:

- Does the paper contain new information adequate to justify publication? Yes

2. Introduction:

- Does the paper clearly explain why the study was necessary? Yes

- Is the research question clear and appropriate? Yes

3. Relationship to literature:

- Does the paper fit the scope of the journal? Yes

- Does the paper adequately understand the relevant literature in the field? No

- Does the paper cite an appropriate range of literature sources? NO

4. Theory, Data and Methodology:

- Is the paper's argument built on an appropriate base of theory? No

- Does the paper utilize an appropriate database? Yes

- Is there enough detail to repeat the study? No

- Are the methods of analysis employed appropriate? Yes

5. Results:

- Are the results presented clearly and accurately? No

- Do the results presented match the methods? No

- Do the authors logically explain the findings? No

6. Discussion and Conclusions:

- Do the authors compare the findings with current findings in the research field? No

- Are the implications of the findings for future research and public policy discussed? No

- Are any contradictory data discussed? Yes

- Are any limitations of the study discussed? Yes

- Are the conclusions supported by the data presented? No

7. Organisation and presentation:

- Is the writing style clear and appropriate to the readership? Yes

- Is the paper well organized? Yes

- Does the content of the paper justify its length? Yes

Revision comments

01. Rewrite the title.

This is a study about socioeconomic inequality in food consumption.

02. Rewrite abstract.

Do not explain the meaning of standard statistical methods in the abstract.

Explain better the results. The text is unclear.

03. Delete the sentence “NOVA this is not an acronym”.

04. Put the ethics statement at the end of the paper.

05. There is no data on obesity.

Rewrite the paper by focusing on socioeconomic inequality in food consumption.

06. Changes in eating habits due to changes in disposable income are a well-known phenomenon (i.e., the so-called generalized Engel’s Law). This literature should be included in the Discussion section. For instance, Nsabimana et al. (2020).

07. Data should be available to other researchers upon request or stored in public databases.

Similar studies not cited in the paper:

Frank, T., Ng, S. W., Lowery, C. M., Thow, A.-M., & Swart, E. C. (2024). Dietary intake of low-income adults in South Africa: ultra-processed food consumption a cause for concern. Public Health Nutrition, 27(1). https://doi.org/10.1017/s1368980023002811

References

Nsabimana, A., Bali Swain, R., Surry, Y., & Ngabitsinze, J. C. (2020). Income and food Engel curves in Rwanda: a household microdata analysis. Agricultural and Food Economics, 8(1). https://doi.org/10.1186/s40100-020-00154-4

Reviewers' comments:

Reviewer's Responses to Questions

**Comments to the Author**

1. Does this manuscript meet PLOS Global Public Health’s publication criteria? Is the manuscript technically sound, and do the data support the conclusions? The manuscript must describe methodologically and ethically rigorous research with conclusions that are appropriately drawn based on the data presented.

Reviewer #1: Yes

Reviewer #2: Yes

2. Has the statistical analysis been performed appropriately and rigorously?

Reviewer #1: Yes

Reviewer #2: Yes

3. Have the authors made all data underlying the findings in their manuscript fully available (please refer to the Data Availability Statement at the start of the manuscript PDF file)?

Reviewer #1: Yes

Reviewer #2: Yes

4. Is the manuscript presented in an intelligible fashion and written in standard English?

Reviewer #1: Yes

Reviewer #2: Yes

5. Review Comments to the Author

Reviewer #1: Major revision comments

01. The title does not fit the manuscript. It suggests that this was an intervention study which is not. The study examines changes in socioeconomic inequalities in food consumption between two surveys and not any intervention to fight overweight and obesity.

02.The main analysis which used Oaxaca decomposition analyzed only UPF product spending which the authors claim is linked to obesity, yet the title mentions both overweight and obesity. Please explain this mismatch.

03. Who was overweight and obese in the study? No definition for this classification has been provided.

04. The authors need to explain and explore the assumption of homogeneity in 5 food consumption patterns among individuals within each household. Although the study only included women aged 15 to 49, most of the food consumption indices are calculated using the household size. Food consumption patter may be different for sex (male vs female) and age, for example infants <6 months should be exclusively breastfed.

05. Table 1 show the major key variables. Most of these variables have not been defined or explained, for example how was rural and urban residence defined? How were the socioeconomic quintiles calculated? Which household assets were included?

Reviewer #2: The manuscript addresses a major public health issue affecting South African population. this is a very useful well conducted study that will go along way in assisting with formulation of workable strategies to curb overweight and obesity in South Africa and beyond. I have a few minor correction that need attention to improving the quality of the manuscript. Pg 5 line 6. The acronym WCBA needs to be explained further. Pg 11 line 15 and elswhere, the use of the abbreviation "i.e." should be discouraged and use the word itself. Page 27 line 1 and elsewhere, organizations such as WHO should include their location when referencing. Pg 27 line 42 and elsewhere, be consistent when writing journal names in the reference list by capitalising first letter in each word of the name or not doing it at all.

6. PLOS authors have the option to publish the peer review history of their article (what does this mean?). If published, this will include your full peer review and any attached files.

**Do you want your identity to be public for this peer review?** For information about this choice, including consent withdrawal, please see our Privacy Policy.

Reviewer #1: **Yes: **Dr Moses Ngari

Reviewer #2: **Yes: **Paul Chelule

---

## [Editor Report · Decision Letter 1]

2 Oct 2024

Explaining socioeconomic inequality in food consumption patterns among households with women of childbearing age in South Africa

PGPH-D-24-00301R1

Dear Dr Mweete D. Nglazi,

We are pleased to inform you that your manuscript 'Explaining socioeconomic inequality in food consumption patterns among households with women of childbearing age in South Africa' has been provisionally accepted for publication in PLOS Global Public Health.

Best regards,

Fabrizio Ferretti, Ph.D.

Guest Editor